# Study on the Durability of the T-Beam Based on Chloride Ion Erosion

**DOI:** 10.3390/ma13071504

**Published:** 2020-03-25

**Authors:** Lifeng Gu, Guowen Yao, Xuanrui Yu, Qiaoyi Li

**Affiliations:** 1State Key Laboratory of Mountain Bridge and Tunnel Engineering, Chongqing Jiaotong University, Chongqing 400074, China; 990020050526@cqjtu.edu.cn; 2School of Civil Engineering, Chongqing Jiaotong University, Chongqing 400074, China; 611190080010@mails.cqjtu.edu.cn (X.Y.); 622180970036@mails.cqjtu.edu.cn (Q.L.)

**Keywords:** T-beam, chloride concentration, numerical simulation, bending capacity, shear capacity

## Abstract

Based on analyzing the bearing capacity of existing T-beam bridges in service, the factors that affect the T-beams cracked by chloride ions mainly include the width and the depth of cracks. Combined with practical engineering examples, a single-piece T-beam model is established to explore the influence of factors such as crack width and crack depth on the T-beams affected by chloride ion erosion through numerical simulation in this paper. In addition, the attenuation models of bending capacity and shear capacity of the T-beam are obtained to analyze the possible failure modes of T-beams with cracks. All of which provides a reference for exploring the effect of crack width and depth on the durability of reinforced concrete members under chloride ion field.

## 1. Introduction

By the end of 2018, the total number of highway bridges in China exceeded 800,000, and the total number of railway bridges exceeded 200,000. However, the disturbance of the external environment and insufficient maintenance during construction have led to serious structure failure, especially reinforced concrete beam bridges, of which the upper structure integrity is poor, making it easier to crack and damage. The specific failure modes can be roughly divided into reinforcement corrosion, concrete carbonization, concrete spalling, etc. In addition, due to the key factors affecting the bridge bearing capacity, such as increased traffic, larger vehicle loads, and adverse surrounding environmental factors, corrosion and damage of reinforced concrete structures in coastal and offshore areas are serious. Numerous projects require a lot of funds for maintenance just shortly after completion, and some are even scrapped in advance. According to the results of the “Corrosion Survey” of bridges in China, the total additional cost caused by the corrosion of steel bars is more than 2 trillion yuan. Scholars have carried out the corresponding experimental researches: Mukhopadhyaya P et al. [1] placed the reinforced concrete specimens in a 5% sodium chloride solution and conducted an experiment at a temperature of 20 °C. After nine months, it was found that the adhesion between the steel bar and the concrete was damaged and significant corrosion occurred on the surface of steel bar, leading to the significant decrease in beam carrying capacity. Luo Shanshan et al. [2] simulated the actual working environment of RC bridges in subtropical coastal areas. After 30 days of salt spray treatment, it was found that the average concentration of chloride ion is 0.228% in the range of 7–14 mm from the steel bar, and 0.127% in the range of 14–20 mm, which was far higher than the initial concentration of the test piece, and the steel bar began to rust. The above study indicates that the damage of chloride ion to reinforced concrete beam cannot be ignored.

T-beams are widely used in urban bridges in China due to their strong bearing capacity. And with the effects of long-term load, a large number of cracks often occur on the surface of T-beams. However, the impact of cracks within limit sizes on the bearing capacity of T-beams is not obvious; T-beams with small cracks are regarded as healthy components of bridges. Initially, the width and depth of cracks were not strictly managed by corresponding code, accordingly, plenty of unsafe bridges have emerged in coastal areas, of which the service lives are less than 10 years [3,4,5].

Studies have shown that the corrosion of steel bars by chloride ions is one of the significant factors weakening the bearing capacity of T-beams. In addition, chloride ions will penetrate into the reinforced concrete T-beams through the growing cracks, when the concentration of chloride ions on the surface reaches the critical concentration, the passivation film of the steel bar will be destroyed, at the meantime, chloride ion reacts readily with the steel bar in humid environment, causing the corrosion phenomenon of steel bar [6,7]. Subsequently, oval pits of different sizes will generate on the surface, resulting in a significant potential difference between the corroded and uncorroded parts, with which the reaction will continue until the steel bar is broken [8,9,10]. The fracture of stirrup leads to the shear capacity deterioration of T-beam while no obvious changes occur in the bending capacity, accordingly, T-beams are prone to shear failure. With the chloride ion gradually penetrating inside the T-beam, the corrosion of the main reinforcement will result in significantly reduction of bending capacity, with which the T-beam is prone to bending failure. The impact of external environment on the bearing capacity has not been taken into consideration of the design code for T-beams at present, and the failure mode of the T-beam is mainly regarded as bending failure, which is inconsistent with the actual situation.

In order to explore the influence of chloride ion on the flexural and shear capacity of T-beams with cracks, Zhao Xiyu et al. [11,12] assumed concrete to be a semi-infinite medium, and analyzed its corrosion pattern through physical model tests, and obtained the diffusion law of chloride ions in concrete and the shear capacity attenuation law of steel bars. Lu Chaohui et al. [13,14] proposed an empirical model for predicting the shear capacity of reinforced concrete beams with physical model tests. Hui Yunling et al. [15,16] analyzed the process of chloride ion erosion in reinforced concrete beams, and concluded that reinforcement would hinder the erosion of chloride ion into concrete. However, the influence of crack sizes on chloride ion diffusion rate is neglected in above researches while various cracks always generate on the beam surface whose width and depth great affect the chloride diffusion rate in engineering [17]. Eiselstein, L.E. et al. [18] found that the corrosion degree is related to the chloride ion concentration and the environment humidity. Through monitoring and studying the environmental impacts under the bridge, Vit Krivy et al. [19] found that, due to the close correlation between corrosion product thickness and corrosion loss, the thickness of the corrosion product can be used to accurately assess the corrosion rate. With consideration of the effects of temperature and time on the coefficient of chloride diffusion, Farahani et al. [20] proposed a model for estimating the service life of RC structures under chloride exposure. Consequently, a T-beam model with cracks is proposed by COMSOL software with numerical simulation analysis in this paper. The length and width of cracks are set according to the measurement of actual engineering to guarantee the accuracy of numerical simulation results. With the comparison of multiple models, we can explore the influence of crack width, depth and other factors on the bending capacity and shear capacity of the T-beam, besides, evolution law of bending and shear capacity over time can be obtained.

## 2. Numerical Simulation Process

An appearance inspection of a T-beam bridge in Hainan Province is shown in Figure 1.

Figure 1 shows that numerous vertical cracks have emerged at the web position of the beam under the coupling effect of load and environment, and the action of dead load from bridge superstructure and live load will cause the continuous derivation of cracks. Additionally, the diffusion rate of chloride ions in cracks will increase simultaneously, which accelerates the corrosion process of reinforced concrete. To simulate the diffusion process of chloride ion in a T-beam with cracks, three finite element models A, B, and C are established using the dilute material transfer module in COMSOL Multiphysics.

Sizes of the T-beam are set according to actual project. the specific size data are shown in Figure 2.

Cracks of the beam body are often located at the web position, as the bottom plate and flange plate of the beam body are not easy to crack. Crack (II) represents the crack in the middle or upper part of the beam that is far from the main rib, as shown in Figure 3a. After penetrating into the beam through crack (II), chloride ions would mainly destroy the passivation film on the stirrup surface, and then affect the shear bearing capacity of the T-beam. Additionally, crack (I) represents the crack near the beam bottom, with which chloride ion concentration on the main tendon surface will increase significantly, greatly increasing the corrosion possibility, and the beam is prone to bending failure.

The main bars are numbered A1–A6, B1–B6, and C1–C6. In addition, three vertical cracks are randomly arranged on the web. The width of the three cracks in model A is 0.3 mm, the depths are 20, 25, and 30 mm, and the lengths are 300, 400, and 500 mm, respectively. The width of the cracks in model B is 0.4 mm, the depths are 20, 25, and 30 mm and the lengths are 300, 400, and 500 mm, respectively. Finally, the width of the cracks in model C is 0.5 mm, with the same crack depths and lengths as cracks in model A and B.

Cracks are numbered L1–L9, and after establishing the whole model with solid elements, mesh the model. In addition, to guarantee the calculation accuracy and the fast calculation speed, a top-down mapping grid is adopted in the model, where the grids are dense near cracks and reinforcement while others are relatively sparse. The maximum unit size is 4 mm and the unit number is 23,457. The concrete surface directly in contact with air is set as the Dirichlet boundary, and the other surfaces are set as Neumann boundaries. The chloride ion concentration field is randomly distributed along the beam in practice. Consequently, it is quite difficult to accurately obtain the chloride ion concentration at various beam positions, as the change trend of the concentration in the environment is always related to the random factors such as humidity and temperature. The chloride ion concentration on the beam surface is based on [21] in this paper, which is 228 mol/m^3^ (only applicable to bridges in offshore areas), and the chloride ion concentration in the beam center is set to 0 without considering chloride impurities in cement and mortar. The FE (Finite element) model is based on Fick’s second law.

With the FE model, the time for the chloride ion to reach the critical concentration on the surface of the steel bar, i.e., the time of depassivation can be obtained, and subsequently the bending capacity and the shear capacity of the steel bar can be calculated. Besides, the adhesive force between the reinforcement and the concrete is not considered, because the chloride ion and the reinforcement are prone to electrochemical reaction in humid environment, as shown in Figure 4.

Once the reaction occurs, corresponding oxidation products will be formed on the surface of the steel bar over time, like red rust (Fe_2_O_3_) when oxygen is sufficient, and black rust (Fe_3_O_4_) when oxygen is insufficient; these products will quickly destroy the adhesion between steel bars and concrete, also known as corrosion expansion. The adhesion between steel and concrete is unreliable. Accordingly, to improve the reliability of the model, the adhesion between steel bars and concrete is not considered in this model.

## 3. COMSOL Calculation Results Analysis

Concretes of different strength levels correspond to different critical chloride ion concentrations. Specific critical chloride ion concentrations are set referring to Table 1.

Large differences exist among the chloride ion diffusion coefficients of the crack locations and non-crack locations, with which the diffusion coefficients of different regions require adjustment. Chloride ion diffusion coefficients are arranged on the locations of substructure surface and web cracks of T-beams, and transfer coefficients of web cracks and non-cracks are adopted according to Equation (1), respectively.
(1)$$D_{\sigma }=\{\begin{array}{c} D_{0}\;\;\;\;\;\;\;\;\;\;\;\;\;\;\;\;\;\;\;\;\;\;\;\;\;\;w>0.1mm\\ D_{m}{\left(\frac{D_{0}}{D_{m}}\right)}^{\frac{100w-30}{70}}\;\;\;\;\;\;\;0.03mm<w<0.1mm\\ D_{m}\;\;\;\;\;\;\;\;\;\;\;\;\;\;\;\;\;\;\;\;\;\;\;\;w\leq 0.03mm \end{array}$$
where D_0_ = 2.03 × 10^−9^ m^2^/s, D_m_ = 1.34 × 10^−11^ m^2^/s, w denotes the crack width. When the crack width is smaller than 0.03 mm, the impact of crack width on the chloride penetration rate is not obvious, the diffusion coefficient of chloride ion is approximately equal to the diffusion value in concrete. However, with the increase of crack width, the diffusion coefficient of chloride ion in the beam also increases. At this point, the amount of chloride ion entering the beam from the crack is much greater than from the concrete. We consider that it is the chloride ion penetrating the beam from the crack that causes damage to the beam.

After model calculation, a main reinforcement surface section and a stirrup surface are proposed to obtain the corresponding surface chloride concentration data of model A, B and C, as shown in Figure 5.

Figure 5 exhibits the results of chloride ion corrosion on T-beams. The outer edge indicates the initial chloride ion concentration, and the chloride ion concentration at the bottom location is the highest followed by the one around cracks. Entering the T-beam through cracks, the corrosion of stirrups by chloride ions will weaken the shear capacity of the T-beam.

In order to analyze the chloride ion concentration on the reinforcement surface to explore the initially corrosion time of steel bars, by setting two-dimensional profiles on both the stirrup surface and the main reinforcement surface, we can obtain three curves of chloride ion concentration, changing over time at the corresponding crack locations of stirrups and main reinforcement, shown in Figure 6. The ordinate means the chloride concentration on the surface of the reinforcement, and the abscissa denotes the corrosion time of chloride ion on the T-beam. The concentration change trend is shown in the blue wireframe in Figure 6; specifically, the chloride ion concentration on the stirrup surface near cracks gradually increase with time. Whereas no significant chloride ion concentration changes happen on the main reinforcement surface at early stages, as shown in the yellow wireframe. In addition, the increase of chloride ion concentration on main reinforcement surface requires enough corrosion time, the reason may lie in the distance between main reinforcement and cracks, with which the early transmission would spend more time. Generally speaking, the growth rate of chloride ion concentration on stirrup surface is greater than the one on the main reinforcement surface.

Input the chloride ion concentration data on main reinforcement surface and stirrup surface of models A, B, and C, respectively. Table 2 shows the chloride ion concentration corresponding to a crack in the model, and the chloride ion concentration on the surface of the steel bar will reach the critical concentration within one year in cracked reinforced concrete beams; subsequently, the steel bar begins to rust.

Considering the most adverse factors of the bearing capacity of T-beam structures, take the maximum chloride ion concentration on the stirrup surface as the chloride ion concentration in the T-beam, as well as the main reinforcement, and we can obtain the depassivation time of stirrup and the main reinforcement under various working conditions, as shown in Table 3.

Table 3 exhibits the depassivation time of stirrup and the main reinforcement. Under chloride ions corrosion, the depassivation time of stirrups is significantly shorter than that of the main reinforcement, and both of which are shorter than one year, indicating that the beam will be quickly attacked by chloride ions once the cracks occur, and the subsequent failure of passivation film further weakens the durability of the beam.

Besides, the depassivation time of stirrups will shorten with the increase of crack width or crack depth; due to which the protective capacity of concrete protective layer on stirrups is gradually weakening as crack width and depth increase, which would lead to shorter transfer time of chloride ion to stirrup surface. However, no significant changes occur on the depassivation time of the longitudinal tensile steel bars as crack width and depth increase, because the chloride ions at the longitudinal tensile steel bars are transmitted through the concrete protective layer, not cracks.

To guarantee the accuracy of the FE model, chloride concentration of various model areas, test results by Scott Thompson Shill [22] and FDTD calculation results are compared together. Figure 7 shows the comparison results: The concentration of different layers in concrete shows an exponential decay trend, and the concentration attenuation trends of the three are similar, which guarantees the model rationality.

## 4. Attenuation Calculation of T-Beam Bending Capacity

Corrosion of chloride ions on stirrups and main reinforcement in the T-beam will weaken the shear bearing capacity and bending bearing capacity of steel bars apparently. As for the calculation method of T-beam bending capacity after corrosion, Shi Qingxuan [23] et al. finished corresponding researches and proposed the calculation method as follows:(2)$$M_{c}=\alpha_{1}f_{c}bx(h_{0}-\frac{x}{2})+\alpha^{\prime }{}_{sc}A^{\prime }{}_{sc}(h_{0}-\alpha^{\prime }{}_{s})$$

Take account of the equilibrium conditions:(3)$$\alpha_{1}f_{c}bx=\alpha_{sc}f_{sc}A_{sc}-\alpha^{\prime }{}_{sc}f^{\prime }{}_{sc}A^{\prime }{}_{sc}$$
where *α*_1_ is the ratio of the stress value of the equivalent rectangular stress diagram to the design value of the axial compressive strength, taken according to code (GB 50010-2010 Code for Design of Concrete Structures in China); *f*_c_ means the axial compressive strength of concrete; *b* denotes the section width, *x* means the height of concrete compression area; *α*_sc_ and *α*’_sc_ are the strength utilization coefficient of corroded reinforcement in compression area and tension area, respectively; *f*_sc_, *f*’_sc_ are nominal yield strength of corroded steel bars in compression and tension areas, respectively; and *A*_sc_, *A*’_sc_ represent the cross-sectional area of corroded reinforcement in the compression area and tension area, respectively.

The strength utilization coefficient of corroded steel bars is the ratio of stress of tensile steel bar under ultimate load to yield strength of tensile steel bar: $\alpha_{s}=\sigma_{s}/f_{y}$, where $\sigma_{s}$ is the stress of tensile steel bar under ultimate load and $f_{y}$ is the yield strength of tensile steel bar. Relationship between the strength utilization coefficient of corroded steel bars and the reinforcement ratio are shown below:(4)$$\alpha_{sc}=\{\begin{array}{cc} 1 & \;\beta_{oc}<0.1mm\\ 1+\frac{x_{c}(0.449-1.822\beta_{oc})}{0.3} & \;0.246mm<\beta_{oc}<0.444mm\\ 1-\frac{x_{c}(0.778-0.634\beta_{oc})}{0.3} & \;\beta_{oc}>0.444mm \end{array}$$

Calculation method of the reinforcement ratio is in accordance with the above reinforced concrete design code.
(5)$$\overset{}{\beta_{oc}}=\frac{f^{\prime }A_{sc}}{f_{c}bh_{0}}$$
where *A*_sc_ is the reinforcement corrosion area; *f’* means the nominal yield strength of corroded reinforcement; *f*_c_ denotes the axial compressive strength of concrete; and *b*, *h*_0_ are section width and effective height, respectively.
(6)$$f^{\prime }=(1-1.196\eta_{s})f$$
where *η_s_* means the corrosion rate of steel section and *f* denotes the yield strength for unrusted steel bars; the corrosion rate calculation method of reinforcement section is shown as follows:(7)$$\eta_{s}=1-{\left(1-\frac{x_{c}}{0.5d}\right)}^{2}$$
where *d* is the reinforcement diameter and *x*_c_ means the reinforcement corrosion depth. In addition, the corrosion depth of steel bar can be obtained by integrating the corrosion rate of steel bar with time, when *λ_cl1_* < 1.8 *λ_cl_*, take *λ_cl1_* = 1.8 *λ_cl_*.
(8)$$\lambda_{cl1}=(4.5-26\lambda_{cl})•\lambda_{cl}\lambda_{cl1}$$

Among which:(9)$$\lambda_{cl}=11.6\times i\times 10^{-3}$$
where *λ_cl_* represents the average corrosion rate of the reinforcement; *λ_cl1_* denotes the steel bar corrosion rate; and *i* means the reinforcement corrosion current density (uA/cm^2^).
(10)$$\ln i=8.617+0.618\ln M_{sl}-\frac{3034}{T+273}-5\times 10^{-3}\rho +\ln m_{cl}$$
where *M_sl_* is the chloride ion concentration on the reinforcement surface (kg/m^3^), taken according to Table 2; *m_cl_* is the local environmental factor; *T* denotes the reinforcement temperature (°C) or ambient temperature; and *ρ* represents the concrete resistivity (KΩ·cm).
(11)$$\rho =k_{p}(1.8-M_{{}_{cl}}^{\mu })+10{\left(RH-1\right)}^{2}+4$$

Coefficient *k_ρ_* = 11.1 when water cement ratio *w/c* = 0.3 ~ 0.4 or C40 ~ C50 and *k_ρ_* = 5.6 when water cement ratio *w/c* = 0.5 ~ 0.6 or C20 ~ C30. *M_cl_* denotes the average value of chloride ion concentration in concrete protective layer (kg/m^3^), when *M_cl_* > 3.6, take *M_cl_* = 3.6 (kg/m^3^). RH represents the environmental relative humidity and *Ms_0_* is the added amount of chloride ion in the preparation of concrete (kg/m^3^).

According to the above formula, we can obtain the bearing capacity downward trend of T-beam after depassivation. Figure 8 shows the relationship between crack depth and the bending capacity of T-beam after one-year depassivation of steel bar, and Figure 9 depicts the relationship between crack width and the bending capacity after one-year depassivation.

Figure 8 depicts that deeper cracks result in lower bending capacity of the T-beam and vice versa. In addition, bending capacity of the T-beam is susceptible to crack width. With the same crack depth, greater crack width leads to lower bending capacity, indicating that crack sizes will directly affect the diffusion rate of chloride ions, and even the bending capacity of beams. Figure 9 illustrates the effect of crack width on bending capacity of T-beams at different depths. Comparing results between the changing trend of bending capacity in Figure 8 and Figure 9 shows that, although both width and depth of crack will affect the bearing capacity of the T-beam, the influence of crack width is obviously greater than that of crack depth.

Furthermore, the corrosion time of chloride ion is also a key factor. The decrease in bending capacity of T-beams is not obvious as the depassivation time is not long. While Figure 10 shows the bending capacity changing trend of T-beam under different crack widths and depths over time, specifically, the bending capacity of T-beam drops with time and the curve roughly follows the trend of linear change.

## 5. Shear Capacity Attenuation Calculation of T-Beam

Corrosion of stirrups in T-beams by chloride ions will also reduce the shear capacity of T-beams significantly. In order to quantify the resistance attenuation of reinforced concrete beam to investigate the shear bearing capacity of the reinforced concrete beam after corrosion, the calculation method of shear bearing capacity is shown below:(12)$$V_{u}=\alpha_{1}\alpha_{2}\alpha_{3}(0.45\times 10^{-3})bh_{0}\sqrt{(2+0.6p)\sqrt{f_{cu,k}}\rho_{sv}f_{sv}}$$
where *V*_u_ is the shear capacity limit of RC beam; *α*_1_ is the influence coefficient of different bending moment, taken as 1.0 when calculating the shear bearing capacity near the fulcrum of the simply supported beam and continuous beam; *α*_2_ means the pre-stress increasing factor, taken as 1 for reinforced concrete flexural members; *α*_3_ represents the influence coefficient of compression flange, for the section of compression flange, take as 1.1; $\rho_{sv}$ is the reinforcement ratio of the longitudinal tensile reinforcement in inclined section; and ρ is the reinforcement ratio of longitudinal tensile reinforcement.
(13)$$\rho_{sv}=\frac{A_{sv}}{bs_{v}}$$
(14)$$\rho =\frac{A_{s}}{bh_{0}}$$
where *A_sv_* is the total cross-sectional area of each corroded stirrup limb within the range of one stirrup spacing *s_v_* along the beam length in inclined section, and *s_v_* is the stirrups spacing along the beam length direction.

### Shear Bearing Capacity Attenuation Calculation Results

According to Equations (12) to (14), and combined with the calculation results of chloride ion transfer concentration by finite element method, we can obtain the current amount and the corrosion rate of steel bars, with which the corrosion amount of steel bars at different times after depassivation can be calculated to study the relationship between the shear bearing capacity of debonding steel bars and the factors such as chloride corrosion time, crack width and depth. The results are shown below.

Figure 11 shows the relationship between crack depth and shear capacity with the same crack width. In general, deeper cracks lead to faster corrosion rates of chloride ion on the T-beam, and shear capacity of stirrups gradually reduced with a relatively slow change trend. Additionally, great differences exist among the steel bar shear capacity with different crack widths, indicating that both crack width and depth have an impact on the shear capacity of steel bars. Figure 12 depicts the influence of crack width on shear capacity of reinforcement. To conclude, crack width is inversely proportional to the shear bearing capacity (i.e., wider cracks result in lower shear bearing capacity), and the rapid reduction rate of shear capacity illustrates that the influence of crack width on stirrup shear bearing capacity is greater than that of crack depth.

Furthermore, the corrosion time of chloride ion on the T-beam will also affect the shear capacity of reinforcement, and part of the results are shown in Figure 13.

Figure 13 describes the change of the T-beam shear capacity over time with different crack depths and widths. Generally speaking, longer corrosion time of chloride ion on the T-beam will result in lower shear capacity. Besides, crack width and depth will also affect the shear capacity of the T-beam, which is consistent with the above analysis results.

## 6. Conclusions

(1)A finite element model was established to simulate the erosion process of T-beams by chloride ions in this paper. The depassivation time and the chloride ion concentration on the steel bar surface can be obtained, which contributes to the analysis of T-beam durability.(2)The changing laws of chloride ion concentration on the steel bar surface over time are obtained by COMSOL numerical simulation software and, through corresponding calculation formula, we can acquire the attenuation trend of bending capacity and shear capacity of the T-beam changing with time, which basically obeys the law of linear change.(3)By changing the cracks’ widths and depths, conclusions that crack width and depth will affect the diffusion rate of chloride ions, and that the influence of crack width on the bearing capacity of T-beams is greater than that of crack depth, can be drawn.

## Figures and Tables

**Figure 1 materials-13-01504-f001:**
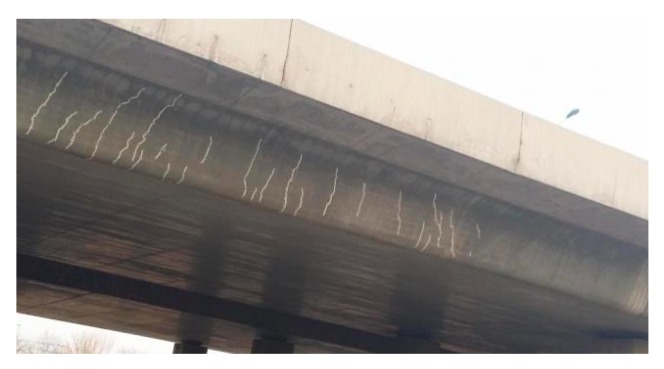
Schematic diagram of T-beam cracks.

**Figure 2 materials-13-01504-f002:**
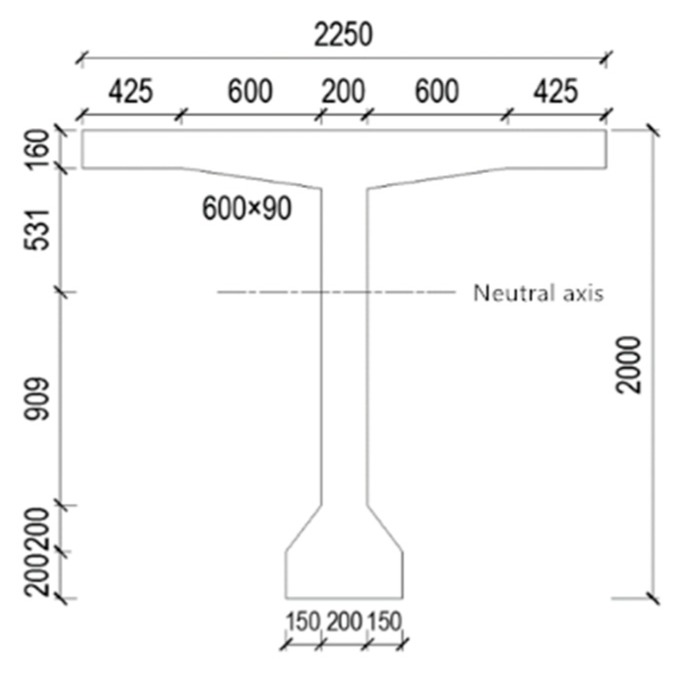
Sketch map of T-beam.

**Figure 3 materials-13-01504-f003:**
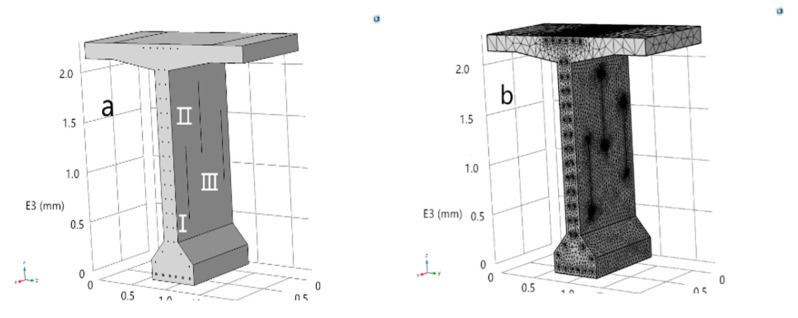
(**a**) The T-beam model; (**b**) schematic diagram of the T-beam model grid.

**Figure 4 materials-13-01504-f004:**
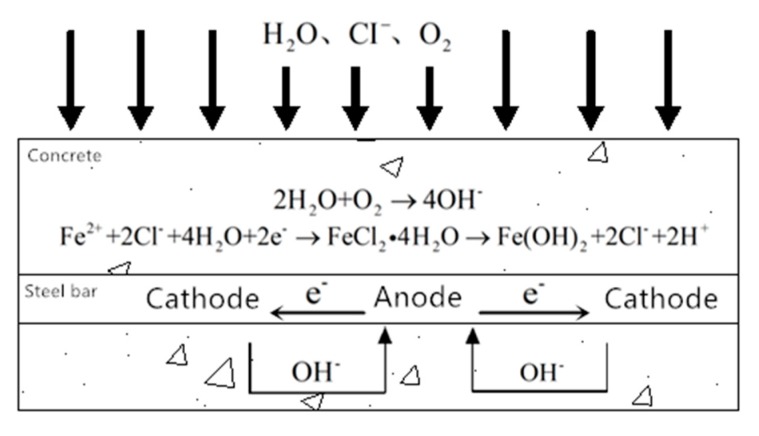
The electrochemical reaction.

**Figure 5 materials-13-01504-f005:**
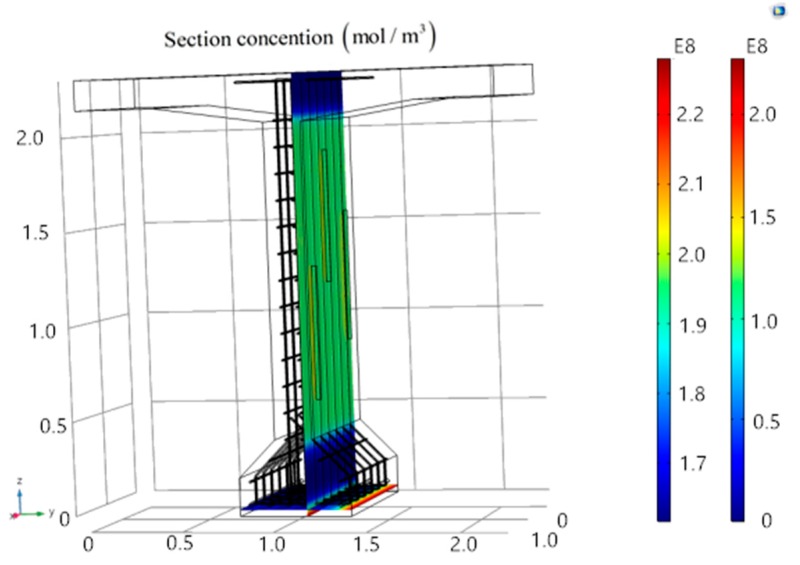
Calculation results of the T-beam model.

**Figure 6 materials-13-01504-f006:**
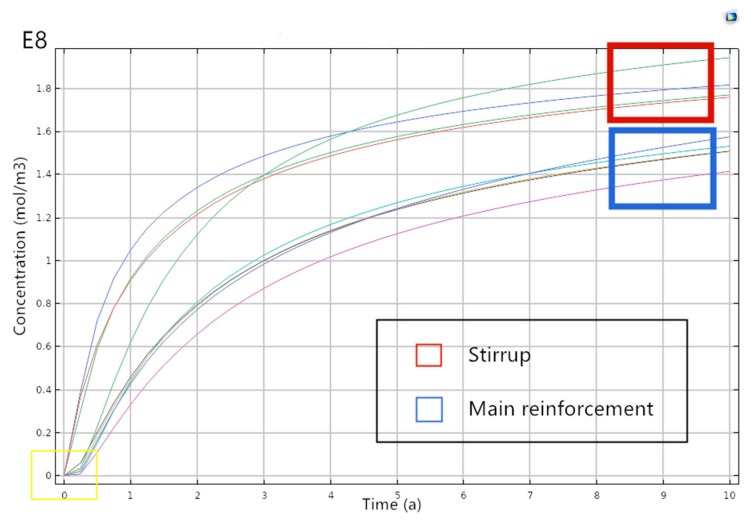
Curve of chloride ion concentration over time.

**Figure 7 materials-13-01504-f007:**
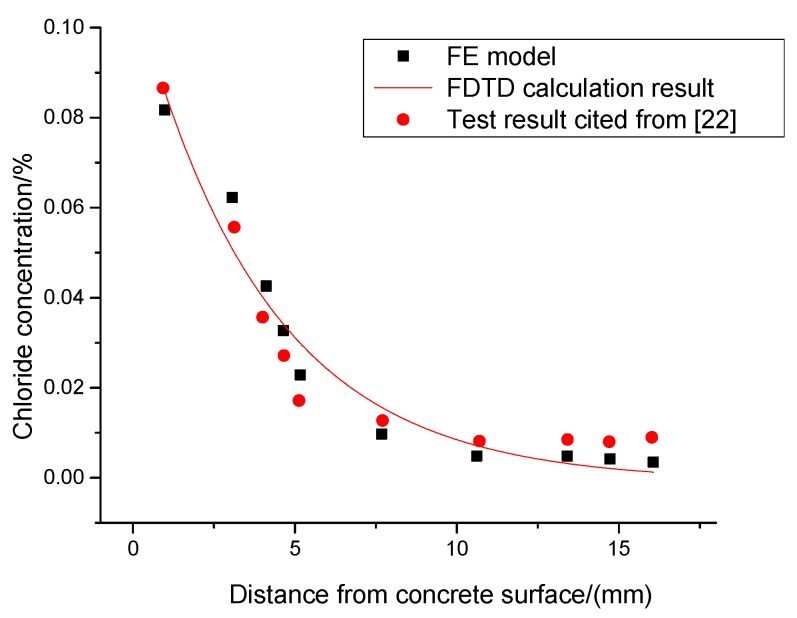
Comparing results.

**Figure 8 materials-13-01504-f008:**
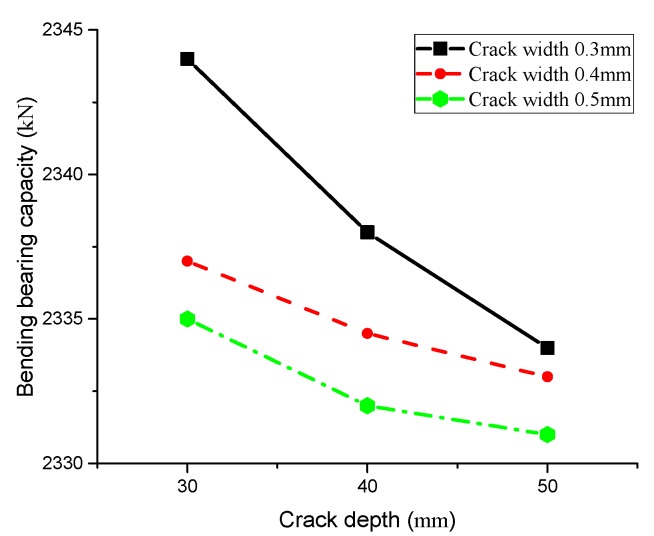
Curve of crack depth and bending capacity.

**Figure 9 materials-13-01504-f009:**
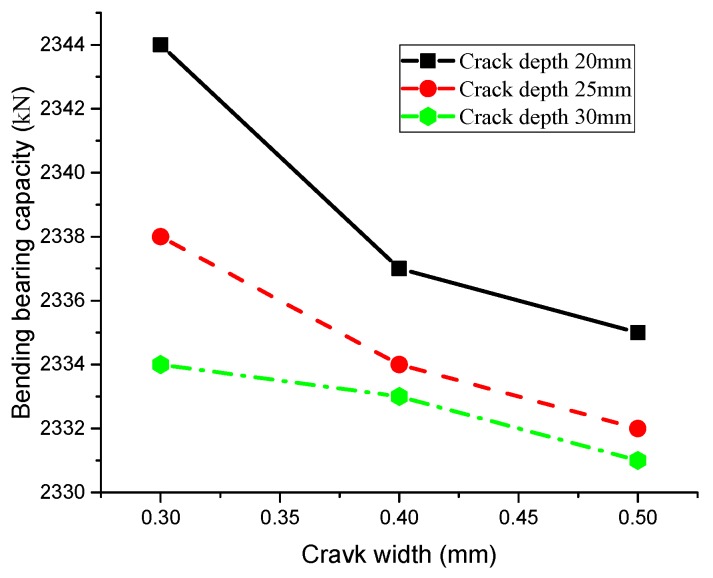
Curve of crack width and flexural capacity.

**Figure 10 materials-13-01504-f010:**
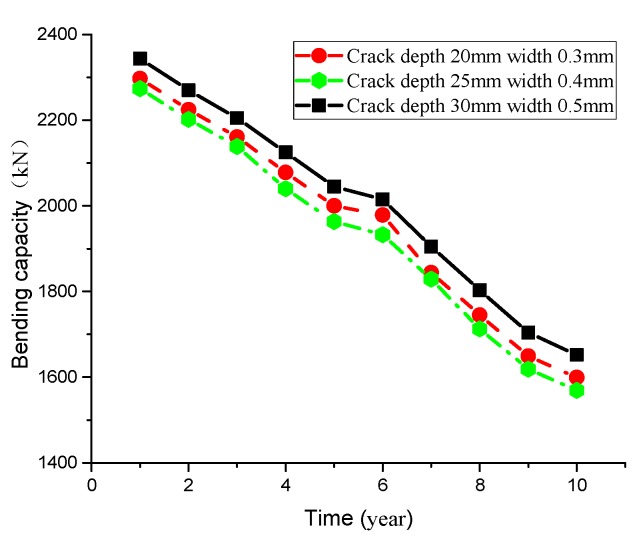
Variation trend of flexural bearing capacity over time.

**Figure 11 materials-13-01504-f011:**
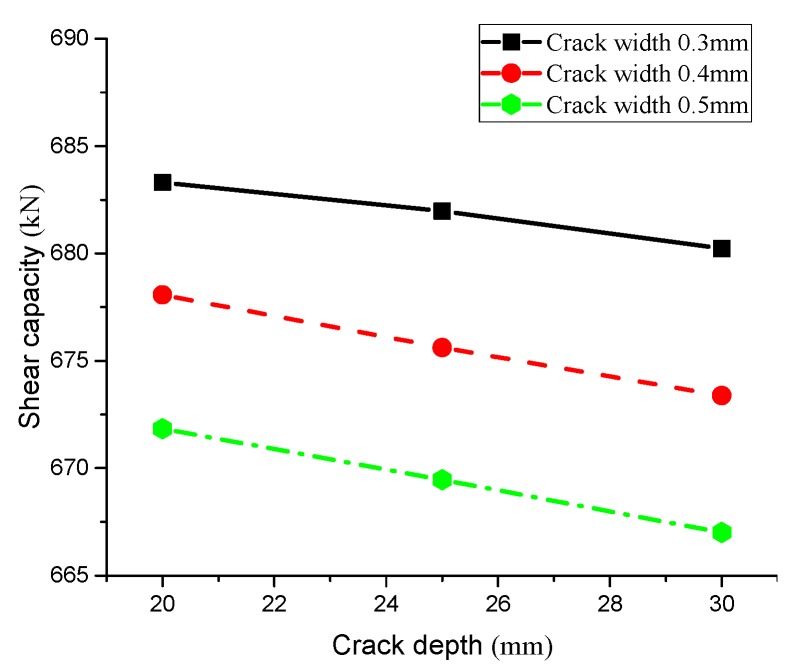
Relationship between crack depth and shear capacity.

**Figure 12 materials-13-01504-f012:**
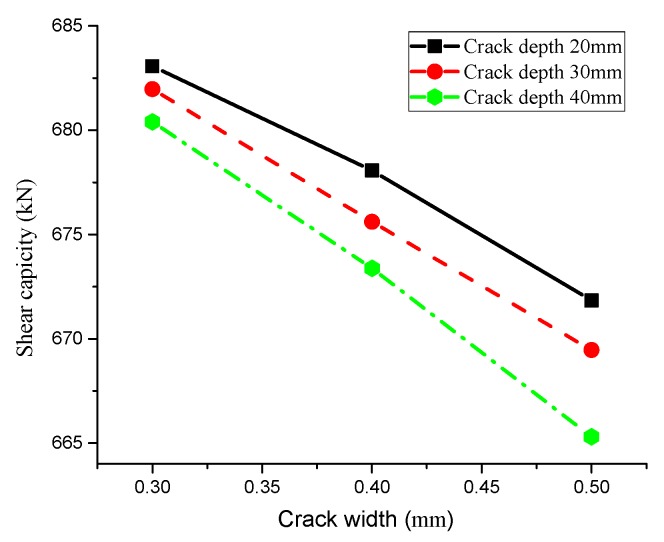
Relationship between crack width and shear capacity.

**Figure 13 materials-13-01504-f013:**
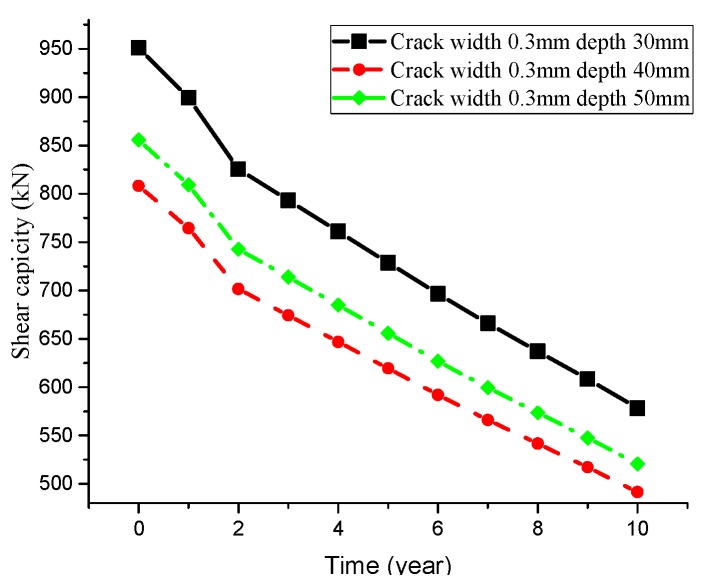
Shear strength attenuation curve.

**Table 1 materials-13-01504-t001:** Critical concentration value of chloride ion in concrete [16].

*f*_cuk_ (MPa)	*M*_cr_ (kg/m^3^)
40	1.4 (0.4%)
30	1.3 (0.37%)
≤ 25	1.2 (0.343%)

**Table 2 materials-13-01504-t002:** Surface concentration values of reinforcement.

Time (year)	Chloride Ion Concentration on Main Reinforcement Surface	Chloride Ion Concentration on Stirrup Surface
1	45.72	92.08
2	80.67	121.78
3	100.56	137.59
4	116.12	149.56
5	126.13	156.85
6	134.10	162.34
7	139.2	167.25
8	143.25	172.08
9	146.56	175.27
10	152.32	177.05

**Table 3 materials-13-01504-t003:** Depassivation time.

Crack Number	Depassivation Time of Stirrup	Depassivation Time of Tensile Reinforcement
L1	0.43	0.68
L2	0.45	0.68
L3	0.34	0.73
L4	0.25	0.67
L5	0.24	0.63
L6	0.22	0.59
L7	0.24	0.62
L8	0.16	0.61
L9	0.25	0.69

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
