# Peer review of "Study on the Durability of the T-Beam Based on Chloride Ion Erosion"

_materials, 2020, doi:10.3390/ma13071504_

Round 1
Reviewer 1 Report
Authors can present how strongly Ions can penetrate into the RC beams. It will help to understand the problem of it. (i.e. after what time of using beams in bridge and for different widhts and dephts of crack the ions will reach the steel bars. )
Its always worth to validate the numerical model by empirical tests or at least compare obtained results with other similar works for example for rectangle shaped beams.
It is also worth to analyse not only cracs in the middle part of the beam but also in the bottom. Have the authors analysed this aspect and which cracks were more destructive for the construction.
The fonts presented in figures are to small. Hardly visible.
Authors could highlited more sufficiently the results of the ions penetration into the RC.
In my opinion the article could be published after some changes.
Reviewer 2 Report
The rationale and validity of individual assumptions employed in the numerical simulation should be demonstrated.
The position of this research in international research trends should be stated by referring essential international papers.
Author Response
Point 1: The rationale and validity of individual assumptions employed in the numerical simulation should be demonstrated.
The position of this research in international research trends should be stated by referring essential international papers.
Response 1: Thanks for your advice. The chloride ion concentration field is randomly distributed along the beam in practice. Consequently, it is quite difficult to accurately obtain the chloride ion concentration at various beam positions, as the change trend of the concentration in the environment is always related to the random factors such as humidity and temperature. The chloride ion concentration on the beam surface is based on reference [21] in this paper, which is 228 mol/m3 (only applicable to bridges in offshore areas), and the chloride ion concentration in the beam center is set to 0 without considering chloride impurities in cement and mortar. The FE model is based on Fick's second law.
And international research trends have been added in the paper, part of which are shown below:
Eiselstein, L.E., et al. [18] found that the corrosion degree is related to the chloride ion concentration and the environment humidity. Through monitoring and studying the environmental impacts under the bridge, Vit Krivy et al. [19] found that due to the close correlation between corrosion product thickness and corrosion loss, the thickness of corrosion product can be used to accurately assess the corrosion rate. With consideration of the effects of temperature and time on the coefficient of chloride diffusion, Farahani et al. [20] proposed a model for estimating the service life of RC structures under chloride exposure.
Reviewer 3 Report
This paper presents findings from tests numerical study aimed to evaluate the durability of T-beam exposed to chloride erosion. The following items are to be addressed before the manuscript can be published:
- The introduction section is well put together, but is significantly short. The authors may use some of the recent works published and listed herein as well as elsewhere to further elaborate on other angles of durability issues in beams used in bridges such as harsh weather and loading conditions.
- https://doi.org/10.1016/j.compositesb.2019.106952
- https://doi.org/10.3390/met7090336
- https://doi.org/10.1155/2017/5819202
- Please add further details on the development of FE model. Information such as element type, mesh size, convergence criteria are needed.
- Was the effect of interface between concrete and steel reinforcement taken into account i.e. bond-slip and how does chloride affect this bond.
- Re-do figure 5. It is not clear.
- Looking at figures 6 and 7, one can see that there is very limited reduction in bearing capacity. Any comments on this? It seems that the effect of chloride is minor.
- How was the FE model validated? The validation process is not very clear.
Round 2
Reviewer 1 Report
In my opinion the manuscript can be accepted for publication
Author Response
Thanks for your advice.
Reviewer 2 Report
- Please discuss the validity of relationship between diffusion coefficient and crack width in Equation (1). How did you obtain this equation?
- What types of boundary condition did you use in chloride ion diffusion analysis?
- Please explain the definition of the strength utilization coefficient of corroded steel bars in Equation (4). How did you obtain this equation?
- Calculation conditions of the numerical simulation in this paper seems limited. Please discuss how the simulation results are related to what can actually happen.
Reviewer 3 Report
Thank you for your efforts.
Author Response
Thanks for your advice.